# Work Function Tuning in Hydrothermally Synthesized Vanadium-Doped MoO$_3$ and Co$_3$O$_4$ Mesostructures for Energy Conversion Devices

**Pietro Dalle Feste** [1,2,†], **Matteo Crisci** [1,3,†], **Federico Barbon** [1], **Francesca Tajoli** [1], **Marco Salerno** [4], **Filippo Drago** [5], **Mirko Prato** [4], **Silvia Gross** [1,2,6,*], **Teresa Gatti** [3,7,*] and **Francesco Lamberti** [1,2,*]

1 Department of Chemical Sciences, University of Padova, Via Marzolo 1, 35131 Padova, Italy; pietro.dallefeste@unipd.it (P.D.F.); matteo.crisci@studenti.unipd.it (M.C.); federico.barbon@unipd.it (F.B.); francesca.tajoli@phd.unipd.it (F.T.)

2 Interdepartmental Centre Giorgio Levi Cases for Energy Economics and Technology, University of Padova, Via Marzolo 9, 35131 Padova, Italy

3 Institute of Physical Chemistry, Justus Liebig University Giessen, 35390 Giessen, Germany

4 Materials Characterization Facility, Italian Institute of Technology, Via Morego 30, 16163 Genova, Italy; marco.salerno@iit.it (M.S.); mirko.prato@iit.it (M.P.)

5 Nanochemistry, Italian Institute of Technology, Via Morego 30, 16163 Genova, Italy; filippo.drago@iit.it

6 Karlsruher Institut für Technologie (KIT), Institut für Technische Chemie und Polymerchemie (ITCP), 76131 Karlsruhe, Germany

7 Center for Materials Research, Justus Liebig University Giessen, 35390 Giessen, Germany

* Correspondence: silvia.gross@unipd.it (S.G.); teresa.gatti@phys.chemie.uni-giessen.de (T.G.); francesco.lamberti@unipd.it (F.L.)

† These two authors contributed equally.

**Abstract:** The wide interest in developing green energy technologies stimulates the scientific community to seek, for devices, new substitute material platforms with a low environmental impact, ease of production and processing and long-term stability. The synthesis of metal oxide (MO) semiconductors fulfils these requirements and efforts are addressed towards optimizing their functional properties through the improvement of charge mobility or energy level alignment. Two MOs have rising perspectives for application in light harvesting devices, mainly for the role of charge selective layers but also as light absorbers, namely MoO$_3$ (an electron blocking layer) and Co$_3$O$_4$ (a small band gap semiconductor). The need to achieve better charge transport has prompted us to explore strategies for the doping of MoO$_3$ and Co$_3$O$_4$ with vanadium (V) ions that, when combined with oxygen in V$_2$O$_5$, produce a high work function MO. We report on subcritical hydrothermal synthesis of V-doped mesostructures of MoO$_3$ and of Co$_3$O$_4$, in which a tight control of the doping is exerted by tuning the relative amounts of reactants. We accomplished a full analytical characterization of these V-doped MOs that unambiguously demonstrates the incorporation of the vanadium ions in the host material, as well as the effects on the optical properties and work function. We foresee a promising future use of these materials as charge selective materials in energy devices based on multilayer structures.

**Keywords:** metal oxide; doping; semiconductor; work function tuning; energy device

## 1. Introduction

Transition metal oxides are a multifaceted and diversified class of inorganic materials whose chemico-physical, structural and functional properties encompass a very wide range of different features [1–4]. From the structural point of view, also depending on their stoichiometry, they display very different structures, such as, for instance, (the most common) NaCl, rutile, corundum, fluorite, spinel and cuprite structures. As far as their electric behavior is concerned, it ranges from superconductors (e.g., high-T$_c$ copper oxides), to good metallic conductors (e.g., V$_2$O$_3$, ReO$_3$) through to semiconductors (e.g., NiO, ZnO,

$TiO_2$, $VO_2$) and they can also display interesting magnetic (e.g., $Fe_3O_4$), and electrochromic (e.g., $WO_3$) characters [2,3]. From the chemical point of view, their properties span the full range, from acidic through amphoteric to basic. These chameleonic features allow these materials to be widely used in commercial products combined to their thermal and chemical stability, ease of processing, and excellent mechanical robustness [5–9].

Focusing on the semiconducting materials, there is flourishing research in the optimization of electronic properties of metal oxides in order to fulfill the stringent requirements of next generation clean energy conversion devices such as solar cells [10].

This is the case, for example, of $MoO_3$ and $Co_3O_4$, that show different electrical properties [11–15] but present one issue: the low mobility of charge carriers. In fact, both oxides are used in photovoltaics for different reasons. First, $MoO_3$ is a n-type semiconductor; however, due to the high band gap (about 3 eV [16]) and a work function similar to that of Au (5.1 eV or higher depending on conditions in which it is measured [17,18]), it is commonly used as hole transporting material (HTM) and/or as blocking layer for hampering photocharges recombination together with other more conductive HTM. This feature increases the overall cost of device production in terms of money and time consumption. In addition, the charge mobility that affects the actual conductivity of the material is strongly dependent on the exact stoichiometry: the electron conductivity in $MoO_3$ is almost zero ($10^{-7}$ S cm$^{-1}$), whereas in the oxygen-poor $MoO_{3-x}$ species, the conductivity can be pushed up by several orders of magnitude, reaching $10^4$ S cm$^{-1}$ in pure $MoO_2$ [19,20]. These results are in agreement with the fact that the atmospheric conditions (mainly either vacuum or ambient) dramatically change the valence/conduction band (VB/CB) values estimation measured with conventional spectroscopic techniques, with a remarkable variation found in the literature that inevitably affects the design of device architectures [17,18]. On the other hand, the spinel $Co_3O_4$ shows a similar low hole mobility (order of $10^{-5}$ S cm$^{-1}$), slightly higher due to the presence of a double electronic direct band gap that allows the oxide to have the proper band gap value (about 1.5 eV) to be directly used as a light absorber and HTM in a pn junction-based next generation all-oxide solar cell [21–24].

As previously mentioned, the charge conductivity is the bottleneck for the real implementation of these oxides in high-performance energy-conversion devices. There are two main strategies used for improving the conductivity of thin films based on these materials: (i) improvement in the quality of the film morphology (i.e., by grain boundaries engineering) and (ii) chemical doping [25] of the metal oxide (MO) constituting the film with heteroatoms to accomplish band engineering instead. Among the different doping elements, specific transition metal ions such as Mn, Fe, Cu, Co, Cr [26–28] are the most promising ones because of the number of oxidation states available that can in principle tune the electronic properties of the host MO. Our choice in this work was addressed towards an aliovalent vanadium (V) substitution, because this metal can be easily oxidized to its pentavalent form in $V_2O_5$, a MO that possesses a work function between 6.4 eV and 7 eV [29]). In this way, other MOs that contains V(V) ions as dopants could be likely p-doped, thus improving their charge mobility, as occurred in some examples in the literature involving vanadium and $MoO_3$ [30].

In the literature, the doping of $Co_3O_4$ has been previously performed using different elements such as Ru, N, La, Sn, Cu, Li, Ag, V and adopting various synthetic/deposition/processing approaches, such as co-precipitation [31], plasma treatment [32], spin coating [33], spray coating [34], dip coating [35], magnetron sputtering [36], thermal decomposition [37], and solvothermal or hydrothermal synthesis [38–41]. This last synthetic methodology is the most valuable among the others because it can be carried out under green conditions (water-based solutions) at low costs (i.e., with simple wet-chemistry laboratory equipment and easy chemical procedures), and typically pursues the fast crystallization of materials already under low temperature subcritical conditions. Moreover, it allows for a good control over reaction conditions (such as the fine tuning of the dopant concentration), while ensuring high yields. However, only a few syntheses are reported in the literature for the preparation of V-doped $Co_3O_4$. Magnetron sputtering [36] is suitable for direct film deposition, but is not usable to obtain

materials to be subsequently processed through a solution-based technique. A case of V-doped $Co_3O_4$ synthesized hydrothermally is reported in the literature for catalytic purposes [41], but the electronic properties are not described, thus remaining hitherto uncharacterized. On the other hand, starting from a theoretical analysis that predicts the possibility of balancing the oxygen vacancies in the lattice, [42] $MoO_3$ has been doped using both chemical and physical deposition techniques such as CVD [43], the sol-gel method [44], the solution combusted method [45], magnetron sputtering [46], and also hydrothermal routes [47,48], similar to $Co_3O_4$. This last procedure is widely applied for realizing doped $Mo_3O_4$ samples with different elements such as Ni [43,45], S [49] and a range of other transition metals [50–52] and in particular, vanadium [30].

In this work, we describe the subcritical hydrothermal synthesis of V-doped $Co_3O_4$ micro-wires and $MoO_3$ micro-lamellae starting from water soluble precursors (followed by a calcination step), with the aim of finely tuning the electronic properties of these MO semiconductors to improve their potential for exploitation in energy-related devices. We also report on the outcomes obtained by implementing a systematic variation in concentration of the introduced dopants (from 1% to 20% at. with respect to the host oxide), which happen to be significantly different for the two investigated MO host matrices, allowing us to draw preliminary conclusions on the arrangement of the vanadium guest species within the two different crystalline lattices. We also show that such a tuning of the doping level in these semiconducting MOs can lead to a parallel tuning of the Fermi level towards the realization of materials with a higher work function compared to the undoped analogues, with interesting perspectives for applications in energy-conversion devices.

## 2. Results and Discussion

In Figure 1, a scheme of the hydrothermal synthesis used to prepare V-doped $MoO_3$ and $Co_3O_4$ mesostructures is presented (experimental details are provided in the Experimental Section). In the first step, the precursors of cobalt or molybdenum and vanadium are dissolved in water together with the vanadium precursor. During the second step, a precipitating agent is added under solution stirring. Next, a Teflon reactor containing the mixture is sealed in a Teflon-lined stainless-steel reactor and heated at 150 °C for a set time to perform the hydrothermal reaction in the third step. The separated powders are finally calcined to yield the final V-doped and undoped (for reference) MOs ($Co_3O_4$:V and $MoO_3$:V).

The samples obtained by this procedure are characterized by powder X-Ray diffraction (P-XRD) to verify the formation of the desired phase (cubic for $Co_3O_4$, space group Fd-3m (227), orthorhombic for $MoO_3$, ($\alpha$-$MoO_3$), space group Pbnm (62)). Figure 2a shows the diffractogram of undoped $Co_3O_4$, confirming the presence of the cubic $Co_3O_4$ phase (PDF 74-1657), with a unit cell parameter $a$ estimated by Pawley refinement (Figure S3a) of $8.0838 \pm 0.0014$ Å (Figure S3a). The reflections located at $2\theta = 19°$, $31.2°$, $36.8°$, $38.6°$, $44.8°$, $55.7°$, $59.5°$ and $65.3°$ are attributed to the planes (1 1 1), (2 2 0), (3 1 1), (2 2 2), (4 0 0), (4 2 2), (5 1 1) and (4 4 0) of $Co_3O_4$, respectively [53]. Diffractograms of V-doped samples (Figure S1 in the Supporting Information—S.I.) show the cubic $Co_3O_4$ pattern and no other phases are detected, nor a shift of the reflections' positions as a consequence of V-doping. An additional phase is detected only at the highest nominal concentration of dopant (i.e., 20% at.), as shown in Figure 2b, where weak reflections appear at $2\theta = 20.3°$, $26.2°$. Specifically, they are ascribable to the lattice planes (0 0 1), (1 1 0) of crystalline orthorhombic $V_2O_5$ (PDF 41-1426) [54], indicating the formation of a second and separated MO phase in the synthetized material when a relatively high amount of V-precursor is introduced in the reaction mixture. The textural characterization with scanning electron microscopy (SEM, Figure 2c) discloses the peculiar morphology of this synthesized $Co_3O_4$, constituted of particles of some tens of nm fused together to generate individual micro-wires (cross-section is in the nanoscale). To the best of our knowledge, a similar hierarchical arrangement of $Co_3O_4$ nanoparticles (of which many examples exist [55,56] also in the form of nanowires [57,58]) into longer wire-like structures has never been reported before.

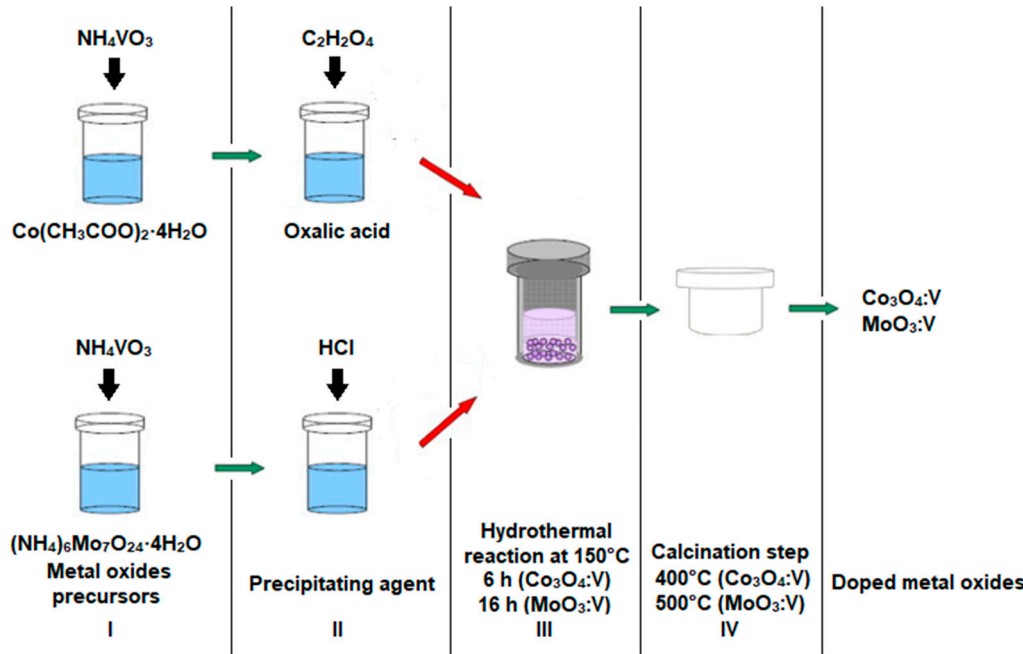

**Figure 1.** Schematic representation of the hydrothermal synthesis procedure used in this work to obtain two metal oxides MOs with vanadium guest species, namely $Co_3O_4$:V and $MoO_3$:V.

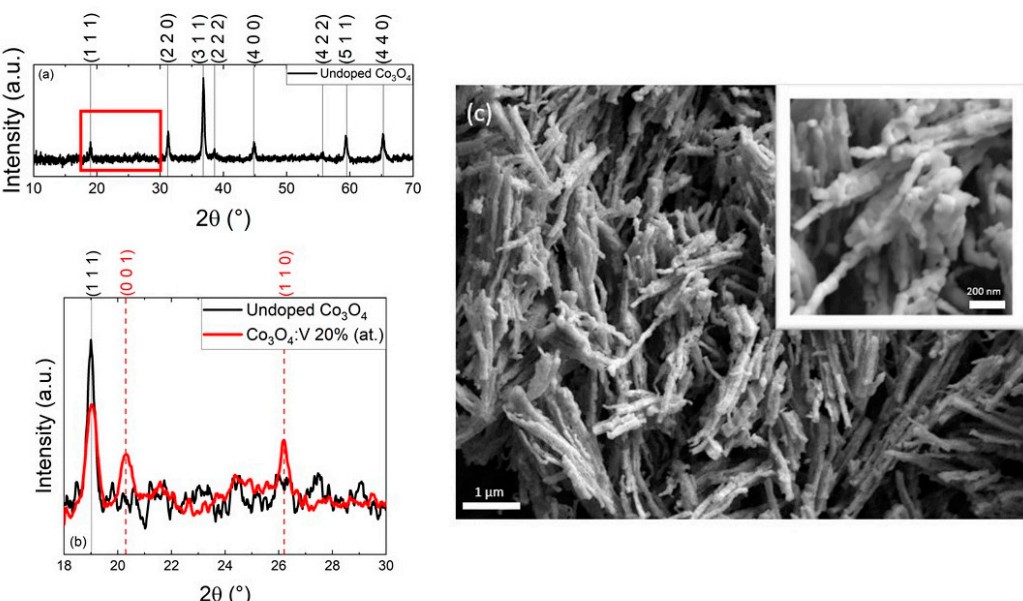

**Figure 2.** (**a**) Powder X-Ray diffraction (P-XRD) diffractogram of hydrothermally synthesized undoped $Co_3O_4$, after the calcination step. (**b**) Zoom on the 18°–30° 2θ range of the same diffractogram and comparison with that of the $Co_3O_4$:V 20% at. sample. (**c**) Field-emission scanning electron microscopy (FE-SEM) image of a typical $Co_3O_4$:V sample (20% at) showing the micro-wire-like morphology. The inset shows a magnified area where the particle-like sub-units of the micro-wires are evidenced.

Figure 3 shows the P-XRD patterns of the undoped $MoO_3$ and $MoO_3$:V 20% at. samples (respectively, in Figure 3a,b). The reflections correspond to those of the orthorhombic $MoO_3$ α-phase (PDF 65-2421) with calculated unit cell parameters of $a = 3.9602 \pm 0.0003$ Å, $b = 13.8540 \pm 0.0011$ Å, $c = 3.6949 \pm 0.0002$ Å estimated by Pawley refinement (Figure S3b). However, this analysis shows that a mixture of orthorhombic (α-$MoO_3$) and monoclinic (β-$MoO_3$, PDF 85-2405) phases of $MoO_3$ is present in the sample. Major diffraction peaks

(with relative lattice planes) corresponding to the desired phase, α-MoO3, can be seen at 12.8°, 23.4°, 25.8°, 27.4°, 33.8°, and 39° 2θ, relative to the planes (0 2 0), (1 1 0), (0 4 0), (0 2 1), (1 0 1) and (0 6 0), respectively. In Figure 3b, two low intensity Bragg peaks at 21.6° and 24.9° 2θ are clearly visible in the V 20 % at. nominal doped diffractogram. From a search and match procedure, these could be related to the presence of $V_9Mo_6O_{40}$ (PDF 34-0527) which is also well fitted in the Pawley refinement (Figure S3c). The absence of such a phase in the samples with a lower percentage of nominal doping could be related to the instrumental detection limit, or the polyoxovanadate could be present in an amorphous state. No shift of the reflections is evident (as can be seen in Figure 3b and from the complete set of diffractograms in Figure S1), so the amount of substitutional $V^{5+}$ ions for $Mo^{6+}$ ions might be negligible. Moreover, the calculated lattice parameters for the $MoO_3$: V 20% at. sample (Figure S3c) differ by less than 0.01 Å with respect to the pure compound, proving that the structural change in the unit cell is very limited. In Figure 3c, an SEM image of a V-doped $MoO_3$ sample is shown, highlighting the presence of lamellar structures of highly variable dimensions in the order of the μm, similar to what was obtained elsewhere in the literature [59]: this structural variability can be due to the calcination step, which promotes the coalescence of the smaller objects with the bigger ones, leading to a non-homogenous distribution of size.

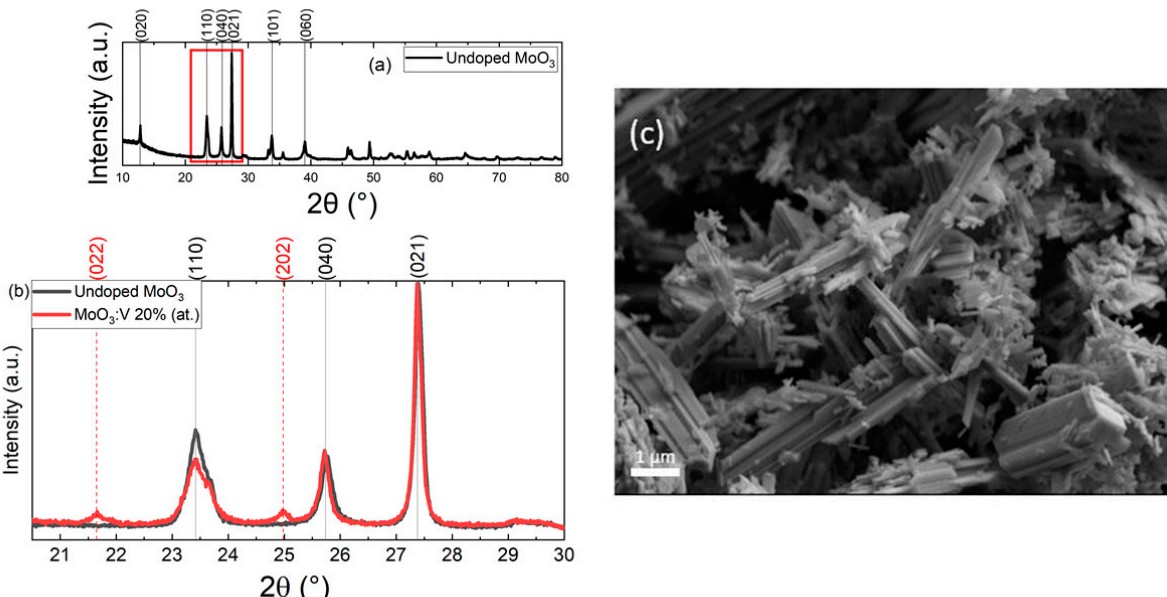

**Figure 3.** (**a**) P-XRD diffractogram of hydrothermally synthesized undoped $MoO_3$. (**b**) Zoom on the 20.5°–30° 2θ range of the same diffractogram and comparison with that of the $MoO_3$:V 20% at. sample. (**c**) SEM image of a typical $MoO_3$:V sample (20% at.) showing the micro-lamellar-like morphology.

The hydrothermally synthesized and subsequently calcined MO samples are optically characterized by diffuse reflectance spectroscopy. The recorded spectra are reported in Figure S2 in the S.I. The spectra of $Co_3O_4$:V samples present two characteristic bands at 700 and 400 nm, attributed to a charge transfer transition from oxygen to the metal center, namely from $O^{2-}$ to $Co^{3+}$ and from the $O^{2-}$ to $Co^{2+}$, respectively [60]. It is worth to mention that, among other MOs, $Co_3O_4$ is particularly sensitive to oxygen due to the simultaneous presence of $Co^{2+}$ and $Co^{3+}$ ions that affect the oxygen content within the lattice. It has been proved that the amount of oxygen can be tuned by the addition of heteroatoms, by treatment with plasma or with different gases [61–63]; however, stabilizing the oxygen vacancy in $Co_3O_4$ remains challenging and specific advanced structural characterization is required in order to characterize it, such as synchrotron extended X-ray absorption fine structure (EXAFS) or X-ray absorption near edge fine structure (XANES).

From the reflectance spectra, a further analysis allows one to determine the band gap energy using the Tauc plot method [64]. This method is based on the relationship existing between light absorbance and band gap energy, described by the equation $(\alpha h\nu)^{\frac{1}{\gamma}} = B(h\nu - E_g)$ [65], where $\alpha$ is the absorption coefficient, $h\nu$ is the incident photon energy, $E_g$ is the energy of the band gap and $\nu$ is a constant; $\gamma$ is equal to 1/2 for direct transitions or 2 for indirect ones. In this case, the reflectance spectra are transformed into the corresponding absorption spectra using the Kubelka–Munk Function ($F(R_\infty)$). In the Tauc plots presented in Figure 4a, the product $(F(R_\infty)h\nu)^{\frac{1}{\gamma}}$ (with $\gamma$ assuming the value of 1/2 for a direct band gap in $Co_3O_4$) is plotted as a function of the incident photon energy, resulting in the determination of two band gaps for the $Co_3O_4$:V samples. The intersection of the linear fit of the Tauc plot with the x-axis, gives the value of the band gap energy, resulting in two band gaps for the $Co_3O_4$:V samples.

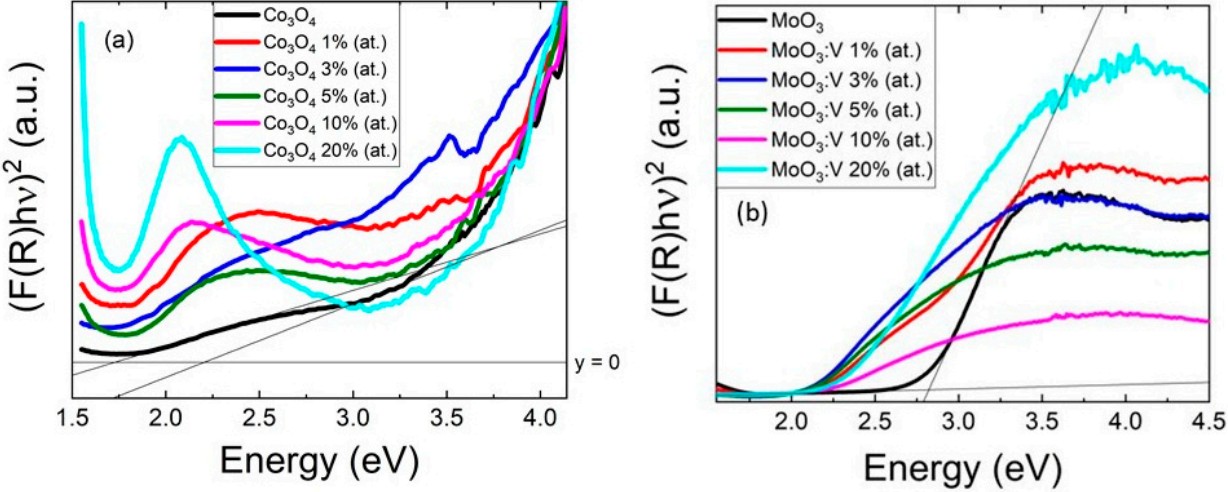

**Figure 4.** Tauc plot analysis for the (**a**) $Co_3O_4$:V and (**b**) $MoO_3$:V samples series. Grey lines indicates the linear fits used to extrapolate the $E_g$ values.

Reflectance measurements and relative band gap calculations were performed for the $MoO_3$:V samples as well (reflectance spectra are reported in Figure S2 of the S.I.). In there, a main band around 375 nm (3.3 eV) can be observed only for the undoped sample, which shifts towards higher wavelengths for the doped samples, following an unclear relationship with the doping concentration. Moreover, a narrow and intense peak is found in the near-infrared (NIR) region at approximately 1080 nm (1.1 eV) that was previously attributed to the intervalence charge transfer $Co^{2+} \rightarrow Co^{3+}$ [66]. The Tauc plot is also built for this second MO series (Figure 4b), with the main difference in the $\gamma$ value used, which in this case is 2 for an indirect band gap semiconductor [65].

The results of all these linear fittings are reported in Table 1. From this analysis, it appears that the bandgap of $Co_3O_4$ remains almost unvaried, with only a slight increase in the sample doped with 20% at. nominal vanadium. For $MoO_3$ instead a progressive decrease in the $E_g$ from the undoped material is evident, up to the 10% at. doped sample, after which a plateau is reached as described by the values reported in Table 1, likely as a consequence of a doping saturation effect.

**Table 1.** Summary of chemical and electrical properties for the series of hydrothermally synthesized undoped and V-doped $Co_3O_4$ and $MoO_3$ samples investigated in the present study.

| Sample | V % [1] | $E_g$ (eV) [2] ($\pm4\%$) | WF (eV) [4] |
|---|---|---|---|
| Undoped $Co_3O_4$ | - | 1.74; 2.24 [3] | 4.8 $\pm$ 0.11 |
| $Co_3O_4$:V (1% at) | <1 | 1.50; 2.20 [3] | 4.75 $\pm$ 0.05 |
| $Co_3O_4$:V (3% at) | <1 | 1.50; 2.20 [3] | 4.71 $\pm$ 0.02 |
| $Co_3O_4$:V (5% at) | <1 | 1.65; 2.14 [3] | 4.71 $\pm$ 0.02 |
| $Co_3O_4$:V (10% at) | <1 | 1.48; 2.35 [3] | 4.81 $\pm$ 0.01 |
| $Co_3O_4$:V (20% at) | 12 | 1.63; 2.62 [3] | 5.15 $\pm$ 0.04 |
| Undoped $MoO_3$ | - | 2.8 | 5.14 $\pm$ 0.05 |
| $MoO_3$:V (1% at) | 1 | 2.3 | 5.34 $\pm$ 0.05 |
| $MoO_3$:V (3% at) | 3 | 2.1 | 5.48 $\pm$ 0.08 |
| $MoO_3$:V (5% at) | 4 | 2.1 | 5.70 $\pm$ 0.02 |
| $MoO_3$:V (10% at) | 6 | 2.2 | 6.10 $\pm$ 0.03 |
| $MoO_3$:V (20% at) | 13 | 2.3 | 6.11 $\pm$ 0.02 |

[1] Actual vanadium atomic concentration as determined by ICP analysis and calculated as: V % = number of V dopant atoms/100 host metal atoms (namely, Mo or Co); [2] $E_g$ extracted from Tauc plot analysis with a $\pm4\%$ associated error; [3] For $Co_3O_4$ two $E_g$ values are extracted; [4] Determined by KPM analysis.

The smooth shape of the Tauc plots at band gap edges reveals the presence of shallow defects within the band gap of the investigated vanadium-doped cobalt oxides (attributed to oxygen vacancies as usually happens in oxides) [67], and a substantial shift of $E_g$ for the vanadium-doped molybdenum ones, suggesting that electronic properties of these MO semiconductors are affected by vanadium inclusion. Therefore, a more detailed electronic characterization is required to understand the effect of the ionic dopant on the energetics of these species, which can be obtained by Kelvin probe microscopy (KPM), together with a precise analytical determination of the actual quantity of dopants present in the lattice, which can be obtained by Inductively Coupled Plasma Optical Emission Spectrometry (ICP-OES) analysis.

Although similar synthetic conditions were adopted for both the V-doped MOs, we can state with good approximation that only $MoO_3$ has been successfully incorporated into the matrix, as shown by ICP analysis, whose results are reported in Table 1. In more detail, significantly lower amounts of vanadium are found within the $Co_3O_4$-based samples with respect to the $MoO_3$-based ones, despite the nominal doping concentrations being similar. This result is in substantial accordance with the trend in $E_g$ values determined from the Tauc plots, in which the band gap does not change in $Co_3O_4$, while it decreases with increasing doping in $MoO_3$, for at least the first three samples (see Figure 4 and Table 1). Furthermore, the ICP analysis highlights that a significative amount of dopant is found already at very low V concentrations in $MoO_3$. However, a shift towards higher diffraction angles of the reflections in the corresponding XRD patterns is not observed, and only at 20 % at. of nominal doping an additional phase is detected (likely $V_9Mo_6O_{40}$). Therefore, with the present data, it is still difficult to understand whether vanadium ions are present as substitutional with respect to molybdenum or in other forms within the sample. On the contrary, the $Co_3O_4$ samples do not show any relevant vanadium incorporation except for the 20% at. specimen, which most likely is a biphasic material, i.e., a mixed $V_2O_5$-$Co_3O_4$ composite, as anticipated from P-XRD (Figure 2b). The reason for this different doping behavior with respect to the vanadium ion dopants can probably be ascribed to the different crystal structures of the Mos, with $\alpha$-$MoO_3$ and $V_2O_5$ being both orthorhombic whereas $Co_3O_4$ has a cubic lattice. In light of these considerations, it appears reasonable to foresee an easier accommodation of the V(V) cations in the structure of $MoO_3$ with respect to that of $Co_3O_4$. On the other hand, we exclude a pure influence of the ionic radii of the cations

on the doping efficiency, because these are comparable for all the cations here considered, with the appropriate coordination number (54 pm for V(V) and 59 pm for Mo(VI) with a coordination number VI, being α-MoO$_3$ constituted by MoO$_6$ distorted octahedra, 58 pm for high spin Co(II) in T$_d$ coordination and 54.5 pm for high spin Co(III) in O$_h$ coordination, with Co$_3$O$_4$ having the typical spinel structure Co$^{II}$[Co$^{III}$]$_2$O$_4$) [68].

KPM analysis provides a value of the work function (WF), which can substantially vary upon effective chemical doping, and thus provides information on the doping character (p- or n-) in the semiconducting materials. The trends in the WF of the MO materials (Figure 5) are highly related to the optical and ICP data: in the Co$_3$O$_4$:V samples series the WF remains practically unchanged (4.7÷4.8 eV) across most doping values except for the highest one, for which a significative variation takes place (from ~4.75 eV up to ~5.15 eV). The increased WF being obtained only at the highest V-doping level in Co$_3$O$_4$ can be related to the presence of the second V$_2$O$_5$ phase formed, which is perhaps located on the external surface of the MO mesostructures, thus largely contributing to the measured value (WF of pure V$_2$O$_5$ should be ~7 eV [69]). On the contrary, for the MoO$_3$ samples, a linear increase in the WF (from 5.14 eV to 6.10 eV) is observed with increasing dopant concentration, for the lowest values thereof, while the WF does not increase further only at the highest concentration (with a 6.11 eV WF value obtained when the nominal amount of vanadium in the MO doubles from 10% at. to 20% at.), thus suggesting also in this case the occurrence of a phase separation within the material as for the case of Co$_3$O$_4$:V 20% (at). These results (with the expected doping concentrations corroborated by ICP data), suggest that only in the case of MoO$_3$ a hypothetical substitutional p-doping of the material is obtained by using the hydrothermal method at low vanadium concentrations (i.e., below 10% at.), whereas for Co$_3$O$_4$ any additional determined p-character can be only attributed to the presence of a V$_2$O$_5$ phase formed in conjunction. However, an interesting two-phase composite material appears with high crystalline content as revealed by P-XRD, which possesses the characteristics of both the constituents (the E$_g$ of pristine Co$_3$O$_4$ and a deeply located Fermi level similar to V$_2$O$_5$), what could be useful for photovoltaic applications.

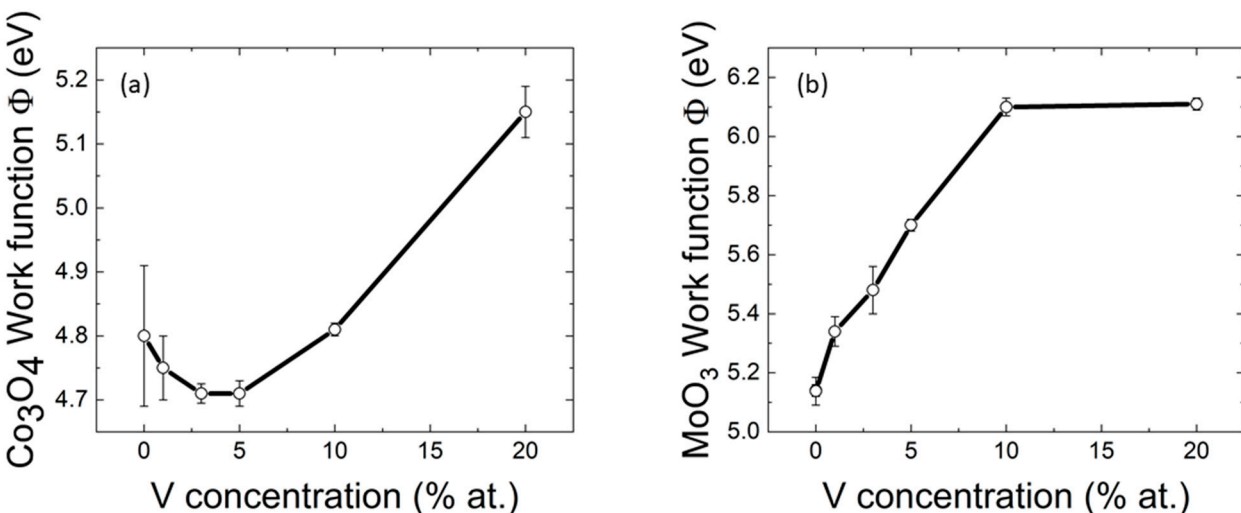

**Figure 5.** Work function values estimated through Kelvin probe microscopy analysis for the (**a**) Co$_3$O$_4$:V and (**b**) MoO$_3$:V samples series.

## 3. Conclusions

In this work, we demonstrated the effect on structural and electrical properties of the incorporation of vanadium ions in MO host materials resorting to a green hydrothermal method that starts from polyoxidometalate or polyoxovanadate and carboxylate precursors. In one case, V ions are incorporated in the MoO$_3$ lattice up to a relatively high atomic percentage (10% at.). We also showed the effect of such an incorporation on the work

function of the MO semiconducting scaffold, which results in a progressive (and almost linear) increase (with respect to the vacuum level), thus allowing us to predict an effective p-doping of the material that enables tuning of the charge transport properties of the materials, whose actual determination remains the subject of future work. In addition, it should be pointed out that the nature of the Mo(VI) substitution with aliovalent extrinsic V(V) cations in the lattice is only hypothesized at the moment, and its confirmation would be obtained through X-ray absorption spectroscopy investigations performed at synchrotron, which is a suitable method to investigate the typology of ion incorporation into doped materials.

On the other hand, we showed that a low band gap $Co_3O_4$ MO semiconductor (also prepared through hydrothermal synthesis) cannot guest V(V) cations into its lattice, although V(V) and Co(II/III) ionic radii are quite similar in dimension, and the modulation of the WF is achievable only when a relatively high concentration of vanadium precursor is added (experimental 12% at.), thus forming a two-phases compound based on $V_2O_5$ and $Co_3O_4$. Although the synthetic goal of this work was mainly demonstrating the possibility of substitutional (or even interstitial) doping of the two considered MO matrices, we will next characterize this Co-V composite further, to better understand its structural features and evaluate its real potential for application in sustainable technologies for conversion of light energy into other useful forms (electricity, solar fuels, but even into light energy storage applications, given the relevance of cobalt oxides in battery-like devices).

## 4. Experimental Section

All reagents and solvents were purchased from Sigma Aldrich (Darmstadt, Germany), if not otherwise specified, and used as received. Powder X-ray diffraction was carried out on a Bruker D8 Advance Plus diffractometer at the PanLab department facility, founded by the MIUR "Dipartimenti di Eccellenza" grantNExuS. Diffraction data were acquired by employing the Cu $K_\alpha$ radiation (Cu anode supplied with 40 kV and a current of 40 mA). A LYNEXEYE XE-T detector with 192 measuring channels in 1D mode was used. The patterns for $Co_3O_4$-based samples were recorded in the range of 10°–70° with a 0.027° (2θ) scan step and 0.7 s per step acquisition time. For the $Co_3O_4$:V 20% at. sample, one additional XRD pattern was recorded in the range of 18°–40° with a 0.027° (2θ) scan step and 2 s per step acquisition time. The XRD patterns of the $MoO_3$-based samples were collected in the range of 10°–80° with a 0.014°(2θ) scan step and 0.3 s acquisition time per step. Fixed divergence slits of 0.50° were used together with Soller slits with a 2.5° aperture.

The crystallographic phase identification was performed by a search and match procedure, using Bruker DIFFRAC.EVA [70] software. The diffractograms were analyzed with the software TOPAS Academic V6 [71] (Bruker AXS). Pawley refinements were carried out by fitting the background with a Chebychev function with ten parameters. The zero-point error or the sample displacement were refined, along with the lattice parameter of the phases and the crystallite size. The shape of the reflections was modeled through the fundamental parameter approach incorporated in the program, separating the instrumental and the sample contributions. Fit indicators $R_{wp}$, $R_{exp}$, and GoF (Goodness of Fit) were used to assess the quality of the refined structural models.

The OriginPro (OriginLab, MA, USA) [72] software was used to elaborate the data.

Scanning electron microscopy images were recorded with a Zeiss SUPRA 35VP instrument. FE-SEM images were taken using a primary beam acceleration voltage of 5.0 kV and a SE2 detector (secondary electrons).

UV-Visible reflectance spectra were acquired on a Cary 5000 spectrophotometer equipped with an integrating sphere.

Kelvin probe microscopy measurements were carried out in ambient air with an atomic force microscope MFP-3D by Asylum Research (Oxford Instruments, UK). A probe ASYELEC.02 was used, with nominal cantilever resonance frequency of 300 kHz, and tip coated with 25 nm thick Ti/Ir film. The scan was two-pass, with surface potential measured during the second pass at elevated height (typically 80 nm). The WF of the tip was first determined, by measuring the electrical surface potential V on highly-oriented pyrolitic

graphite (HOPG), whose work function was assumed to be 4.6 eV. Measurements were carried out on a thick (bulk) layer of sample powder, pressed to form a solid pellet, finding consistent values, within the estimated uncertainty ($\pm$100 mV). The obtained value resulted from averaging the means from $n$ = 4 images of typical 5 $\mu$m scan size, in different sample locations.

Elemental analysis was carried out via Inductively Couple Plasma Optical Emission Spectrometry (ICP-OES), with an iCAP 7600 DUO (Thermo Fisher Scientific). Samples were weighted and digested in a single overnight step, in the case of $Co_3O_4$-based samples, in a flask with aqua regia; $MoO_3$-based samples were instead digested in a two-step process, first with HF and then with aqua regia. The digested samples were diluted to 10 mL using ultrapure Milli-Q water and filtered using a 0.45 $\mu$m PTFE filter before the analysis. ICP-OES was used to determine the relative amount of vanadium with respect to the metal in the MO sample. Values were evaluated using the Qtegra software (Thermofisher).

*Hydrothermal Synthesis of V-Doped $Co_3O_4$ and V-Doped $MoO_3$*

For a typical hydrothermal synthesis of V-doped $Co_3O_4$ and V-doped $MoO_3$, adapting procedures reported in the literature [41,73], a given amount of metal precursor are dissolved in deionized water (10 mL). The amounts of cobalt acetate tetrahydrate (Carlo Erba, Milano, Italy, pur. >99%) and ammonium eptamolybdate tetrahydrate (Alfa Aesar, Kandel, Germany, pur. 99%) are respectively 0.3736 g and 0.58 g. At 0.15 M, solutions of cobalt acetate are added 0.0018, 0.0053, 0.0088, 0.0175, 0.0351 g of ammonium metavanadate (pur. 99%), with a molar ratio to the cobalt precursor of 0.01, 0.03. 0.05, 0.1 and 0.2, respectively, to have in the mixtures' different atomic ratios between the vanadium and cobalt precursor (1%, 3%, 5%, 10%, 20% at.). In a similar way, to the 0.05 M solutions of ammonium eptamolybdate 0.0746, 0.2237, 0.3728, 0.6984, 1.3968 g of ammonium metavanadate are added, with a molar ratio to the molybdenum precursor of 0.0746, 0.2237, 0.3728, 0.6984 and 1.3968, respectively, to achieve different atomic ratios between the vanadium and molybdenum precursor (1%, 3%, 5%, 10%, 20% at.). At the cobalt precursor solution, 0.135 g of oxalic acid (pur. >99%) with a molar ratio of 1:1 with respect to cobalt acetate are added, followed by stirring for 30 min. The molybdenum precursor solution is instead stirred for 20 min, then 275 $\mu$L of HCl 37% v/v are added (0.005 molar ratio to ammonium molybdate), and next the solution is stirred for other 20 min. Each solution is poured in a 23 mL A255AC PTFE cup (Parr Instrument Company, IL, USA).

**Supplementary Materials:** The following are available online at https://www.mdpi.com/2076 -3417/11/5/2016/s1, Figure S1: Diffractograms of (a) $Co_3O_4$-related compounds and (b) $MoO_3$-doped samples. Figure S2. Reflectance spectra of (a) $Co_3O_4$-related compounds and (b) $MoO_3$-doped samples used for Tauc plot analysis. Figure S3. Diffractograms and corresponding Pawley refinements of (a) $Co_3O_4$ (Rwp = 9.27 %), (b) $MoO_3$ (Rwp = 9.41 %) and (c) $MoO_3$:V 20% (at.) (Rwp = 9.47 %) with unit cell parameters of: a = 3.9617 $\pm$ 0.0004 Å, b = 13.8660 $\pm$ 0.0013 Å, c = 3.6934 $\pm$ 0.0003 Å.

**Author Contributions:** Conceptualization, S.G., T.G. and F.L.; methodology, S.G., T.G. and F.L.; validation, S.G., T.G. and F.L.; formal analysis, T.G. and F.L.; investigation, M.C., P.D.F., F.B., F.T., F.D., M.S., M.P., S.G., T.G. and F.L.; resources, M.S., M.P., F.T., S.G., T.G.; data curation, T.G. and F.L.; writing—original draft preparation, M.C., P.D.F., T.G. and F.L.; writing-review and editing P.D.F., M.C., F.T., F.B., S.G., T.G. and F.L.; visualization, F.L. and T.G.; supervision, S.G., T.G. and F.L.; project administration, S.G., T.G. and F.L.; funding acquisition, S.G., T.G. and F.L. All authors have read and agreed to the published version of the manuscript.

**Funding:** P.D.F., F.L. and S.G. gratefully acknowledge the Interdepartmental Centre Giorgio Levi Cases for Energy Economics and Technology of the University of Padova for financial support within the project AMON-RA. S.G. thanks the Department of Excellence Project "NEXUS" for providing the P-XRD equipment and laboratory spaces. T.G. and M.C. thank the European Commission for the H2020 FET-PROACTIVE-EIC-07-2020 project LIGHT-CAP (project number 101017821) and the Verband der Chemischen Industrie e.V. for the "Fonds der Chemischen Industrie". F.T. and F.B. gratefully acknowledge for funding the PhD course in Molecular Sciences and the PhD Course in Science and Engineering of Materials and Nanostructures, University of Padova, Italy, respectively.

**Institutional Review Board Statement:** Not applicable.

**Informed Consent Statement:** Not applicable.

**Data Availability Statement:** Not applicable.

**Conflicts of Interest:** The authors declare no conflict of interest.

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
