# Peer review of "Work Function Tuning in Hydrothermally Synthesized Vanadium-Doped MoO3 and Co3O4 Mesostructures for Energy Conversion Devices"

_applsci, doi:10.3390/app11052016_

Round 1

Reviewer 1 Report

see attached

Reviewer 2 Report

In this study, the authors report on the influence of V doping on the structural features, optical bandgap, and work function of MoO3 and Co3O4 meso-structures synthesized by hydrothermal method. There are no novel ideas or significant conclusions were presented in this manuscript. Moreover, physical mechanisms of some results and discussion are lack.

Comments

  1. Description of the motivation of this work is not clear.
  2. Some latest research results should be mentioned and cited in the section of introduction instead of outdated or earlier papers.
  3. The effect of V doping level on the structural features should be clearly explanation.
  4. Please comment the role of oxygen vacancy of the samples on the optical bandgap.
  5. The optical bandgap and work function of the metal oxide meso-structures do not compare with other works.
  6. Characters are too small in some figures.

Reviewer 3 Report

Reviewers report:

The manuscript „Work function tuning in hydrothermally synthesized vanadium-doped MoO3 and Co3O4 mesostructures for energy conversion devices“ by Pietro Dalle Feste, Matteo Crisci, Federico Barbon, Marco Salerno, Filippo Drago, Mirko Prato, Silvia Gross, Teresa Gatti, Francesco Lamberti is devoted to hydrothermal synthesis of V-doped Co3O4 micro-wires and MoO3 micro-lamellae starting from water soluble precursors. The authors present an interesting study on the dependence of chemical and electrical properties of Co3O4 and MoO3 on vanadium concentration, which can be useful for development of light harvesting devices. In general I find the work is interesting and worth to publish in Applied Sciences.

Reviewer 4 Report

The work of Feste et al., reports on the hydrothermally synthesized vanadium-doped MoO3 and Co3O4 mesostructures. The work will be interesting for the readership but the authors need to address some important issues.

  1. Are there any previous reports on MoO3 and Co3O4 microwires in the literature? The cross-section of the microwires is nanoscale. The authors should add this information to the manuscript.
  2. Fig. 1. The authors should add the time for the calcination step. What was the heating rate?
  3. What is the refined zero-point of the diffractometer? Is it parameter considered when plotting Fig. 3b?
  4. What are the hkl indices of the shifted peaks and how they are related to the corresponding atomic planes of isomorphous substitution (Fig. 3b)?
  5. The details of the hydrothermal synthesis should be duly described: amounts of reagents, molar ratios, purity of reagents, and etc.

Round 2

Reviewer 2 Report

I recommend the actual chemical composition, V and O, should be reported in this work. You may perform the XPS or EDS to obtain experimental results.

Author Response

We understand the Reviewer´s request for more analytical data like XPS or EDX, however we point out that ICP measurements already present in the submitted version of the manuscript are a more complete elemental analysis characterization and they provide information on the materials bulk, while XPS or EDX are well-known to be rather surface-sensitive.
Indeed, with a ppm-scale precision, we were able to carry out a thorough speciation of the doped-compounds and to correlate very well the nominal doping concentration used during synthesis with the experimental amount of vanadium present in the resulting sample. For this reason, at this stage, we consider useless to improve the analytical characterization with further analyses.

Reviewer 4 Report

Please include the volume of the PTFE cup for the synthesis.

Author Response

The cup is 23 mL. The information is added to the text